# Comparison of life history parameters of two different genetic clusters of *Bemisia tabaci* MED (Hemiptera: Aleyrodidae) through single and cross mating

**Yujeong Park**[1], **Young-gyun Park**[1], **Joon-Ho Lee**[1,2]*

**1** Entomology Program, Department of Agricultural Biotechnology, Seoul National University, Seoul, Republic of Korea, **2** Research Institute for Agriculture and Life Sciences, Seoul National University, Seoul, Republic of Korea

* jh7lee@snu.ac.kr

**Data Availability Statement:** The data contained within the manuscript and Supporting Information files.

## Abstract

*Bemisia tabaci* Mediterranean (Gennadius) (Hemiptera: Aleyrodidae) is an economically important insect pest worldwide. Previously, we have reported that most *B. tabaci* Mediterranean (MED) populations occurring in greenhouse tomatoes in Korea have been displaced from well-differentiated two genetic clusters (C1 and C2) to one (C2) during one-year period. To elucidate factors responsible for this phenomenon, we compared life history parameters of these two different genetic clusters through single and cross mating experiments on two different host plants, cucumber and tobacco, at 26°C. Intrinsic rate of increase ($r$), finite rate of increase ($\lambda$), and net reproductive rate ($R_o$) were significantly higher in the dominating cluster (C2) (0.247, 1.280, and 192.402, respectively on cucumber; 0.226, 1.253, and 133.792, respectively on tobacco) than in the other cluster (C1) (0.149, 1.161, and 50.539, respectively on cucumber; 0.145, 1.156, and 53.332, respectively on tobacco). Overall performances of cross mating groups, C2fC1m (C2 female × C1 male) and C1fC2m (C1 female × C2 male), were in-between those of C2 and C1, with C2fC1m performing better than C1fC2m. Thus, maternal inheritance appeared to be significantly associated with their life history parameters, with partial involvement of paternal inheritance. Our results demonstrated that the rapid displacement of genetic clusters of *B. tabaci* MED populations was clearly associated with differences in their life history parameters.

## Introduction

The sweet potato whitefly, *Bemisia tabaci* (Gennadius) (Hemiptera: Aleyrodidae), causes significant economic damage to major vegetables, fruits, and ornamental crops worldwide [1–3]. In Korea, *B. tabaci* MED (Mediterranean or biotype Q) is currently predominant in most regions, whereas *B. tabaci* MEAM1 (Middle East-Asia Mininor 1) and *B. tabaci* JpL are only present in a few regions [4, 5]. Specifically, the *B. tabaci* MEAM1 is found in some

**Funding:** This research was funded by grants from the Rural Development Administration (project no. PJ01194804). Also, this study was partially supported by the Brain Korea 21 Plus.

**Competing interests:** The authors have declared that no competing interests exist.

agricultural-producing region and JpL (*Lonicera japonica*) is only present in regions with host plants such as the Japanese honeysuckle (*Lonicera japonica* Thunb).

Previously, we have reported that there are two clusters of *B. tabaci* MED populations in greenhouse tomatoes and that their genetic clusters have been displaced into one genetic cluster in most regions [6]. We hypothesized that the dominating genetic cluster (cluster 2) population, because of higher fitness, could efficiently compete out the other cluster (cluster 1) which was prevalent at the beginning. Potentially different insecticide resistance of these genetic clusters, if any, might be also partly involved in genetic cluster change. Plant virus transmission rates and endosymbionts can also affect host's biology and physiology, thus being able to change genetic cluster [7, 8]. Similar phenomenon has been reported previously in Australia [9] and China [10]. However, there have been no follow-up studies that delve into causes. Life table analysis is considered as one of the most effective analytical tools to evaluate life history parameters of insects [11] because life table parameters provide comprehensive understanding of fitness of insect species [12–15]. More specifically, intrinsic rate of increase (*r*) is a basic parameter for describing population traits [16].

The objective of this study was to provide more evidence for the change in compositions of genetic cluster that resulted in dominance of one genetic cluster of *B. tabaci* MED in Korea. To test our hypothesis that differences in fitness between two genetic clusters contributed significantly to this change, we compared life history parameters of two different genetic clusters of *B. tabaci* MED on two different host plants, cucumber and tobacco, through single and cross mating.

## Materials and methods

### *B. tabaci* MED cultures and plants

We used two different representative genetic cluster populations of *B. tabaci* MED cluster 1 and cluster 2, collected from tomato greenhouses in Pyeongtaek and Sejong, respectively, in Korea in 2018 [6]. We confirmed genetic structures of these populations according to the following procedures. PCR primers were used to amplify microsatellite DNA loci 11, 53 [17], 68, 145, 177 [18], BT4, BT159 [19], and Bem23 [20] using individual gDNAs of *B. tabaci* MED as templates. PCR reaction conditions followed the protocol by Dalmon et al. [18]. PCR products were analyzed using an ABI 3730xl (Applied Biosystems Inc., Foster, CA, USA) at NICEM (Seoul, Korea). Then 1 µl PCR product was diluted with 8.5 µl of Hi-Di formamide (Applied Biosystems Inc.) and 0.5 µl Genescan ROX-500 size standard (Applied Biosystems Inc.). These genetic data were analyzed using GENEMAPPER v.3.7 (Applied Biosystems Inc.), GenAlEx v.6.5 [21], STRUCTURE v.2.3.2 [22], and STRUCTURE HARVESTER Web v.0.6.93 [23].

Host plants used in this study were cucumber (*Cucumis sativus* L.) and tobacco (*Nicotiana tabacum* L.). This is because *B. tabaci* prefers plants with pubescent leaves for oviposition and feeding [24, 25]. These two species are among the most preferred host plants of *B. tabaci* [26]. Both *B. tabaci* MED populations were separately maintained on both cucumber and tobacco plants under the same experimental conditions. *Bemisia tabaci* colonies were reared in cages (40 × 40 × 40 cm) at 26 ± 1°C with relative humidity (RH) of 50 ± 10% and a photoperiod of 14:10 (L:D) h. These colonies served as stock colonies for experiments. The purity of each culture was monitored for every generation by microsatellite analysis. After ten generations of rearing, *B. tabaci* colonies were used for experiments.

### Life table experiments

Life table experiments and analyses were conducted following Maia et al. [27, 28]. Data collection was made from the onset of oviposition of adults until completion of development of their progeny. Followings are our experimental procedures.

To obtain newly emerged virgin adults of *B. tabaci* (< 12-h-old) [29–32], plant leaves with pupae (late 4th instar nymphs with red eyes) were excised from stock colonies of two genetic clusters. The cut of leaf petioles was maintained on a moistened pad until adult emergence. The sex of newly emerged adults was determined under a stereomicroscope (× 200). These adults were separated by sex and placed into insect breeding dishes (10 cm in diameter and 4.2 cm in height) (SPL Life sciences, Pocheon, Korea) before the experiments were initiated.

Life table experiments were conducted for single and cross mating groups between two different genetic clusters of *B. tabaci* MED on two different host plants, cucumber and tobacco (Table 1). All experiments were conducted at 26 ± 1˚C, 50 ± 10% RH, and a photoperiod of L: D (14:10) h in an incubator. Preparation of single and cross mating groups was made using the 'single-pair mating' method [33, 34]. For single-pair mating, we used one female and two male adults of *B. tabaci* in each replicate to assure successful copulation. Each treatment had 30 pairs of *B. tabaci* MED adults. All pairs of *B. tabaci* adults were placed separately on a leaf disc (5 cm in diameter) which was placed on a moistened pad on the bottom of an insect breeding dish (5 cm in diameter and 1.5 cm in height) (SPL Life sciences, Pocheon, Korea). Adults were transferred onto fresh leaf discs in new insect breeding dishes using brushes (Brush 320 Series No. 1, Hwahong, Hwaseong, Korea) every two days. Dead male adults were replaced from colonies. Oviposition and post-oviposition periods, fecundity, and longevity of female adults were observed and counted daily until they died. The survival of offspring for each treatment group was checked for all progeny of individual female adults every two days until they died or became adults. Emerged *B. tabaci* adults were counted and their sex was identified under a stereomicroscope (× 200). Since examination for progeny was made for each female adult with 30 adults for each treatment group, survival rate and sex ratio of all offspring were calculated for each treatment group with 30 replications. To observe developmental period from egg to adult for offspring in each treatment group, a total of 60 eggs were randomly selected among the above described progeny of each group. To ascertain representation of proper progeny of each group, three to five eggs were selected over various randomly allocated dates. Marking was made on lids of insect breeding dishes to identify selected eggs with a permanent marker pen (Name pen X, Monami Co. Ltd, Yong-in, Korea). Their development period was observed daily until they died or became adults. The pad on the bottom of an insect breeding dish was wetted with distilled water using pipette tips every day to maintain healthy leaves.

## Proportion of genetic cluster

To characterize the genetic cluster of each treatment group (i.e., single and cross mating), a total of 20 female individuals from each treatment group were examined using previously described microsatellite analysis procedure. We used a burn-in of 60,000 Markov Chain Monte Carlo (MCMC) steps and a burn-in period of 600,000. We used an ancestry model allowing for admixture and correlated allele frequency among treatments. Log-likelihood

**Table 1. Single and cross mating groups between cluster 1 (C1) and cluster 2 (C2) of *B. tabaci* MED.**

| Host plant | Treatment | Culture type | Mating method | Crosses |
|---|---|---|---|---|
| Cucumber/Tobacco | C1 | Single cluster | Single | C1 (1f × 2m) |
| | C2 | | | C2 (1f × 2m) |
| | C1fC2m | Mixed cluster | Cross | C1 (1f) × C2 (2m) |
| | C2fC1m | | | C2 (1f) × C1 (2m) |

f, female; m, male

estimates were calculated for $K = 1$ to 10 with ten replicates of each. Structure Harvester analysis was performed to detect the likelihood of the number of occurring clusters among individuals of *B. tabaci* MED.

## Body weight and length of adult *B. tabaci*

Body weight and length were measured for 100 female and 100 male adults of *B. tabaci* selected randomly from each treatment group. Adults were frozen. Their body weights and lengths were measured. The body length was measured from the top of the head to the end of the abdomen using a Leica Application Suite X program (Leica Microsystems, Inc., Buffalo Grove, IL, USA). The body weight was measured using a BM-22 microbalance (A&D Co. Ltd., Tokyo, Japan) with 10 individuals as a group.

## Statistical analysis

A two-way analysis of variance (ANOVA) was conducted to determine effects of clusters and host plants on female adult longevity, fecundity, oviposition period, adult body weight, adult body length, offspring's sex ratio, and offspring's survival rate using PROC ANOVA in SAS [33]. PROC GLM in SAS [35] was used for development period of offspring because of different sample sizes among treatments. Mean separation was conducted by Tukey's studentized range test at $p < 0.05$.

## Life table analysis

Fertility life table analysis and jackknife estimation were conducted using the R program (R Development Core Team, 2019) of Maia et al. [28]. Required data for the analysis were the number, longevity, and daily fecundity of female adults from the parent, and the development period, survivorship, and sex ratio from the offspring. Age-specific survival rate ($l_x$) and fecundity ($m_x$) were calculated as follows:

$$l_x = SURV \times \frac{NSF_x}{NF}$$

$$m_x = NEGG_x \times SR$$

Cumulative survival estimation comprises survival of the offspring multiplied by the survival during adult stage which is the number of survived females up to time $x$ ($NSF_x$) and the initial number of females for each treatment group ($NF$). It is necessary to calculate the number of eggs laid at each pivotal age ($NEGG_x$) by the sex ratio of offspring ($SR$) [27]. To calculate the pivotal age (female adult age plus 0.5), average developmental period of the offspring was used [27, 28]. Jackknife estimation and Tukey's studentized range test for population parameters were conducted for all treatment groups for both host plants.Population parameters were as follows [27]:

The intrinsic rate of increase ($r$)

$$\sum_{x=0}^{\infty} e^{-rx} l_x m_x = 1$$

The finite rate of increase ($\lambda$)

$$\lambda = e^r$$

The net reproductive rate ($R_o$)

$$R_0 = \sum_{x=0}^{\infty} l_x m_x$$

The mean generation time ($T$)

$$T = (\ln R_0 / r)$$

## Results

### Proportion of genetic cluster in experimental *B. tabaci* MED groups

In C1 and C1fC2m groups, cluster 1 was dominant. By contrast, cluster 2 was dominant in C2 and C2fC1m groups (Table 2). In single mating, the ratio of the cluster 1 and 2 was over 90 and 70% in C1 and C2, respectively. In cross mating, the cluster 1 and 2 ratio was over 70% in C1fC2m and C2fC1m, respectively. The genetic cluster proportion of each treatment group showed similar pattern on cucumber and tobacco (Fig 1). The genetic diversity indices obtained from all the eight different microsatellite loci of *B. tabaci MED* screened are given in S1 Table.

### Life history parameters

Fecundity, longevity, ovipostion period, survival rate, sex ratio, development period, body weight, and body length of *B. tabaci* MED were significantly different among genetic clusters and between host plants. An interaction effect was also found between genetic cluster and host plants for some characteristics such as fecundity, survival rate, and sex ratio of offspring (S2 Table).

Overall, biological characteristics of *B. tabaci* MED were significantly superior in C2, the lowest in C1, and those of mixed mating groups were in-between. Maternal inheritance was significantly associated with their life history parameters, with partial involvement of paternal inheritance. Total fecundity was the highest for C2 (292.8 ± 2.31 and 244.9 ± 2.29 eggs on cucumber and tobacco, respectively) (mean ± SE), followed by that for C2fC1m, C1fC2m, and C1 on both host plants (Table 3). Female longevity was significantly longest for C2fC1m followed by that for C2 and C1fC2m. The survival rate of offspring (egg to adult) was rather similar among genetic cluster groups (Table 4). Sex ratio (female %) was distinctively higher in C2. It was the lowest in C1. Those of mixed mating groups were in-between. The developmental period (female + male, female, and male) on both host plants from short to long was in the

**Table 2. The proportion of membership according to Bayesian clustering method for two clusters in each treatment group of *B. tabaci* (n = 20).**

| Host plant | Treatment | Inferred Clusters | |
|---|---|---|---|
| | | Cluster 1 | Cluster 2 |
| Cucumber | C1 | 0.968 | 0.032 |
| | C2 | 0.258 | 0.742 |
| | C1fC2m | 0.756 | 0.244 |
| | C2fC1m | 0.157 | 0.843 |
| Tobacco | C1 | 0.968 | 0.032 |
| | C2 | 0.202 | 0.798 |
| | C1fC2m | 0.749 | 0.251 |
| | C2fC1m | 0.166 | 0.834 |

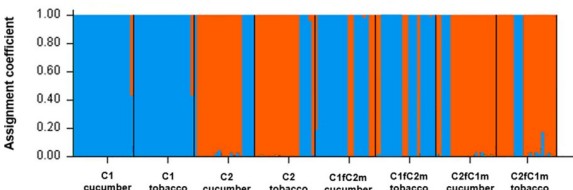

**Fig 1. Scatter plot of STRUCTURE results reporting proportional each treatment of *B. tabaci*.** Each treatment is represented by a vertical line with different colors representing probabilities assigned to each of the genetic clusters. Cluster 1 and Cluster 2 are shown in blue and orange, respectively.

following order: C2, C2fC1m, C1fC2m, and C1 (Table 5). Adult body weight and body length were in the following order: C2 > C2fC1m > C1fC2m > C1 (Table 6).

Overall, C2 outperformed other groups regarding life history characteristics on both host plants (Table 7). Intrinsic rate of increase, finite rate of increase, and net reproductive rate (0.247, 1.280, and 192.402, respectively, on cucumber; 0.226, 1.253, and 133.792, respectively, on tobacco) of C2 were distinctively higher than those of C1 (0.149, 1.161, and 50.539, respectively, on cucumber; 0.145, 1.156, and 53.332, respectively, on tobacco). In cross mating, C2fC1m (0.210, 1.234, and 129.912, respectively, on cucumber; 0.196, 1.216, and 96.196, respectively, on tobacco) outperformed C1fC2m (0.172, 1.188, and 64.292, respectively, on cucumber; 0.168, 1.183, and 57.392, respectively, on tobacco). Intrinsic rate of increase value, finite rate of increase, and net reproductive rate were the highest in C2, followed by those in C2fC1m, C1fC2m, and C1 groups for both host plants.

## Discussion

This study compared life history parameters between two genetically different populations of *B. tabaci* MED based on fertility life table analysis for the first time. Life table parameters of *B. tabaci* have been previously reported regarding different putative species, host plants, or temperatures [36–45]. These studies were conducted to determine the effect and correlation of various conditions. The comparison of these studies demonstrated that the correlation between life table parameters of *B. tabaci* and diverse environments are influenced high.

Different from these previous studies, our life table study was focused on genetically different populations of *B. tabaci* MED to elucidate if difference in life history parameters of

**Table 3. Total fecundity, daily fecundity, longevity, oviposition period, and post-oviposition period (mean ± S.E.) of female *B. tabaci* (n = 30).**

| Host plant | Treatment | Total fecundity | Daily fecundity | Longevity | Oviposition period | Post-oviposition period |
|---|---|---|---|---|---|---|
| Cucumber | C1 | 167.7 ± 3.79dF* | 6.9 ± 0.16bC | 24.6 ± 0.88cC | 23.3 ± 0.75cC | 2.3 ± 0.23bC |
| | C2 | 292.8 ± 2.31aA | 9.6 ± 0.12aA | 30.5 ± 0.35bB | 29.1 ± 0.33bB | 2.4 ± 0.16bC |
| | C1fC2m | 187.3 ± 5.61cE | 6.1 ± 0.13cD | 30.8 ± 0.92bB | 28.7 ± 0.95bB | 3.2 ± 0.47bBC |
| | C2fC1m | 271.8 ± 1.9bB | 7.1 ± 0.10bC | 38.6 ± 0.43aA | 34.1 ± 0.47aA | 5.6 ± 0.46aA |
| Tobacco | C1 | 152.5 ± 1.96dG | 6.2 ± 0.12bD | 24.8 ± 0.58cC | 23.2 ± 0.52cC | 2.6 ± 0.18bC |
| | C2 | 244.9 ± 2.29aC | 7.8 ± 0.08aB | 31.5 ± 0.29bB | 30.0 ± 0.24bB | 2.5 ± 0.18bC |
| | C1fC2m | 174.4 ± 2.53cEF | 5.9 ± 0.14bD | 29.8 ± 0.70bB | 27.9 ± 0.63bB | 3.0 ± 0.21aBC |
| | C2fC1m | 201.5 ± 2.01bD | 5.3 ± 0.07cE | 38.2 ± 0.31aA | 35.4 ± 0.30aA | 3.9 ± 0.20aB |

*Means followed by the same letter (lower case letter, comparison among genetic clusters within a host plant; capital case letter, comparison among genetic clusters throughout both host plants) within a column are not significantly different at α = 0.05, Tukey's studentized range test.

**Table 4. Survival rate and sex ratio (mean ± S.E.) in offspring of *B. tabaci*.**

| Host plant | Treatment | Survival rate of offspring (%) | Sex ratio (%) |
|---|---|---|---|
| Cucumber | C1 | 81.4 ± 1.16aAB* | 38.8 ± 0.90dF |
| | | (4063/5032)** | (1575/4063)*** |
| | C2 | 82.2 ± 0.53aAB | 80.2 ± 0.38aA |
| | | (7220/8785) | (5791/7220) |
| | C1fC2m | 83.3 ± 1.01aAB | 42.2 ± 0.61cE |
| | | (4653/5637) | (1959/4653) |
| | C2fC1m | 80.2 ± 0.76aB | 59.7 ± 0.47bC |
| | | (6529/8154) | (3895/6529) |
| Tobacco | C1 | 83.1 ± 1.00abAB | 41.7 ± 1.27cEF |
| | | (3794/4574) | (1576/3794) |
| | C2 | 81.3 ± 0.66bB | 67.4 ± 0.61aB |
| | | (5964/7348) | (4023/5964) |
| | C1fC2m | 75.8 ± 1.10cC | 42.4 ± 0.61cE |
| | | (3947/5249) | (1670/3947) |
| | C2fC1m | 85.2 ± 0.84aA | 56.3 ± 0.45bD |
| | | (5143/6048) | (2892/5143) |

*Means followed by the same letter (lower case letter, comparison among genetic clusters within a host plant; capital case letter, comparison among genetic clusters throughout both host plants) within a column are not significantly different at α = 0.05, Tukey's studentized range test following arcsine transformation for proportions.
**(survived number / initial number)
***(female number / total adult number)

**Table 5. Developmental period (mean ± S.E.) of *B. tabaci*.**

| Host plant | Treatment | Developmental period | | |
|---|---|---|---|---|
| | | Female + Male | Female | Male |
| | | (n) | (n) | (n) |
| Cucumber | C1 | 21.0 ± 0.17aA* | 19.7 ± 0.12aB | 21.6 ± 0.14aA |
| | | (44) | (15) | (29) |
| | C2 | 15.1 ± 0.07dD | 14.9 ± 0.05dF | 15.9 ± 0.10dD |
| | | (50) | (40) | (10) |
| | C1fC2m | 17.9 ± 0.16bB | 16.8 ± 0.14bC | 18.6 ± 0.13bB |
| | | (46) | (19) | (27) |
| | C2fC1m | 16.5 ± 0.09cC | 16.1 ± 0.07cE | 17.2 ± 0.09cC |
| | | (47) | (29) | (18) |
| Tobacco | C1 | 21.3 ± 0.14aA | 20.5 ± 0.14aA | 21.7 ± 0.15aA |
| | | (42) | (14) | (28) |
| | C2 | 15.3 ± 0.07dD | 15.0 ± 0.04cF | 16.0 ± 0.00dD |
| | | (49) | (35) | (14) |
| | C1fC2m | 17.9 ± 0.18bB | 16.7 ± 0.11bCD | 18.7 ± 0.15bB |
| | | (45) | (18) | (27) |
| | C2fC1m | 16.8 ± 0.12cC | 16.3 ± 0.09bDE | 17.6 ± 0.14cC |
| | | (47) | (29) | (18) |

*Means followed by the same letter (lower case letter, comparison among genetic clusters within a host plant; capital case letter, comparison among genetic clusters throughout both host plants) within a column are not significantly different at α = 0.05, Tukey's studentized range test.

**Table 6. Comparison of body weight and body length (mean ± S.E.) of *B. tabaci*.**

| Host plant | Treatment | Body weight (mg) | | Body length (mm) | |
|---|---|---|---|---|---|
| | | Female | Male | Female | Male |
| Cucumber | C1 | 0.255 ± 0.0061cD* | 0.217 ± 0.0025bBC | 0.666 ± 0.0037dD | 0.514 ± 0.0040dD |
| | C2 | 0.319 ± 0.0010aA | 0.290 ± 0.0017aA | 0.802 ± 0.002aA | 0.682 ± 0.0034aA |
| | C1fC2m | 0.297 ± 0.0052bBC | 0.222 ± 0.0049bB | 0.761 ± 0.0045cC | 0.594 ± 0.0074cC |
| | C2fC1m | 0.312 ± 0.0022abAB | 0.282 ± 0.0032aA | 0.786 ± 0.0030bB | 0.648 ± 0.0045bB |
| Tobacco | C1 | 0.241 ± 0.0054cD | 0.208 ± 0.0023bC | 0.661 ± 0.0033dD | 0.506 ± 0.0038dD |
| | C2 | 0.318 ± 0.0012aA | 0.287 ± 0.0016aA | 0.801 ± 0.0028aA | 0.681 ± 0.0032aA |
| | C1fC2m | 0.290 ± 0.0043bC | 0.220 ± 0.0039bBC | 0.761 ± 0.0038cC | 0.594 ± 0.0052cC |
| | C2fC1m | 0.313 ± 0.0029aAB | 0.279 ± 0.0025aA | 0.782 ± 0.0031bB | 0.634 ± 0.0049bB |

*Means followed by the same letter (lower case letter, comparison among genetic clusters within a host plant; capital case letter, comparison among genetic clusters throughout both host plants) within a column are not significantly different at α = 0.05, Tukey's studentized range test.

different genetic clustered populations might be responsible for rapid displacement to one genetic cluster of *B. tabaci* MED in Korea.

Overall, genetic cluster 2 (C2) of *B. tabaci* MED outperformed genetic cluster 1 (C1) for various aspects of life history parameters through both single mating and cross mating (C2 and C2fC1m vs. C1 and C1fC2m) experiments on both host plants, cucumber and tobacco. These results confirmed that the competitive ability of cluster 2 population was significantly higher than that of cluster 1 regardless of host plant species, indicating that the rapid displacement of genetic clusters of *B. tabaci* MED in Korea populations might be highly related to their different life history parameters.

Fecundity was the highest in C2, followed by that in C2fC1m, C1fC2m, and C1. The same trend was observed for sex ratio, body weight, and body length. The development period was the shortest in C2, followed by that in C2fC1m, C1fC2m, and C1. Since these biological parameters were apparently associated with life history parameters, life table parameters also showed the same pattern. Biological, life history parameters, and clusters proportion of *B. tabaci* MED appeared to be mainly associated with maternal inheritance. To some extent, paternal inheritance was also associated with these parameters. This trend was supported by proportions of genetic clusters in four single and cross mating genetic cluster groups determined by individual-based STRUCTURE analysis (Fig 1 and Table 2). Such genetic inheritance characteristics

**Table 7. Estimates (mean ± S.E.) of life table parameters of *B. tabaci*.**

| Host plant | Treatment | $r$ | $\lambda$ | $R_0$ | $T$ |
|---|---|---|---|---|---|
| Cucumber | C1 | 0.149 ± 0.0006dG* | 1.161 ± 0.0007dG | 50.539 ± 1.3619dE | 26.253 ± 0.2220aB |
| | C2 | 0.247 ± 0.0007aA | 1.280 ± 0.0010aA | 192.402 ± 1.3592aA | 21.300 ± 0.0605dE |
| | C1fC2m | 0.172 ± 0.0008cE | 1.188 ± 0.0010cE | 64.292 ± 2.5132cD | 24.226 ± 0.2735bC |
| | C2fC1m | 0.210 ± 0.0005bC | 1.234 ± 0.0006bC | 129.912 ± 0.9356bB | 23.136 ± 0.0617cD |
| Tobacco | C1 | 0.145 ± 0.0006dG | 1.156 ± 0.0007dG | 53.332 ± 0.7421cE | 27.365 ± 0.1071aA |
| | C2 | 0.226 ± 0.0005aB | 1.253 ± 0.0006aB | 133.792 ± 1.1781aB | 21.680 ± 0.0476dE |
| | C1fC2m | 0.168 ± 0.0012cF | 1.183 ± 0.0014cF | 57.392 ± 0.7397cE | 24.145 ± 0.1895bC |
| | C2fC1m | 0.196 ± 0.0007bD | 1.216 ± 0.0008bD | 96.196 ± 0.9831bC | 23.329 ± 0.0681cD |

*Means followed by the same letter (lower case letter, comparison among genetic clusters within a host plant; capital case letter, comparison among genetic clusters throughout both host plants) within a column are not significantly different at α = 0.05, Tukey's studentized range test after jackknife estimates.

$r$, intrinsic rate of increase; $\lambda$, finite rate of increase; $R_0$, net reproductive rate; and $T$, mean generation time

could accelerate the prevalence of cluster 2 populations. Beside the life history traits, plant virus transmission rates and endosymbionts are known to affect the biology and physiology of their host [46–49]. Therefore, we investigated whether the presence of absence of tomato yellow leaf curl virus (TYLCV), a representative virus mediated by *B. tabaci*, and *Wolbachia* was associated with the genetic cluster changes. However, neither TYLCV nor *Wolbachia* was associated with the changed genetic clusters (S3 and S4 Tables). In this study, we did not examine the potential difference in insecticide resistance of two genetic clusters of *B. tabaci* MED. Insecticide resistance might also play a role in the prevalence of genetic cluster 2 [50]. Further study is needed to clarify this.

In conclusion, this study provided a strong evidence that genetic cluster 2 of *B. tabaci* MED had significantly superior life history parameters than cluster 1. Thus, the rapid displacement of genetic clusters in *B. tabaci* MED populations is strongly related to their different life history parameters. Further study is needed to determine potential difference in insecticide resistance between these two genetic clusters of *B. tabaci* MED.

## Supporting information

**S1 Table. Genetic diversity of the *B. tabaci* MED treatments.**
(DOCX)

**S2 Table. Results of two-way ANOVA for testing effects of cluster and host plant on biological parameters, body weight, and body length of *B. tabaci*.**
(DOCX)

**S3 Table. Tomato yellow leaf curl virus detected of *B. tabaci* MED populations in Korea from 2016 to 2018.**
(DOCX)

**S4 Table. *Wolbachia* detected of *B. tabaci* MED populations in Korea from 2016 to 2018.**
(DOCX)

## Author Contributions

**Conceptualization:** Yujeong Park, Young-gyun Park, Joon-Ho Lee.

**Data curation:** Yujeong Park, Young-gyun Park, Joon-Ho Lee.

**Formal analysis:** Yujeong Park, Young-gyun Park.

**Investigation:** Yujeong Park.

**Methodology:** Yujeong Park, Young-gyun Park, Joon-Ho Lee.

**Resources:** Yujeong Park.

**Software:** Young-gyun Park.

**Supervision:** Joon-Ho Lee.

**Validation:** Joon-Ho Lee.

**Visualization:** Yujeong Park.

**Writing – original draft:** Yujeong Park.

**Writing – review & editing:** Yujeong Park, Joon-Ho Lee.

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
