## [Decision Letter · Decision Letter 0]

8 Jan 2021

PONE-D-20-28992

Comparison of life history characteristics of two different genetic clusters of Bemisia tabaci MED (Hemiptera: Aleyrodidae)

PLOS ONE

Dear Dr. Lee,

Thank you for submitting your manuscript to PLOS ONE. After careful consideration, we feel that it has merit but does not fully meet PLOS ONE’s publication criteria as it currently stands. Therefore, we invite you to submit a revised version of the manuscript that addresses the points raised during the review process.

To reach a level acceptable for publication in PLOS ONE, major revisions, mainly concerning the MATERIALS and METHODS and DISCUSSION sections should be made by the authors. There are some unclear and/or confusing technical and scientific issues and missing info (from previous studies linked to the same topic being investigated here) and references throughout the manuscript, therefore authors are suggested to provide all necessary info and to better present their findings to improve the scientific quality of the manuscript. Please see and follow Reviewers' comments stated below.

We look forward to receiving your revised manuscript.

Kind regards,

Ramzi Mansour

Academic Editor

PLOS ONE

Journal Requirements:

2.We suggest you thoroughly copyedit your manuscript for language usage, spelling, and grammar. If you do not know anyone who can help you do this, you may wish to consider employing a professional scientific editing service.  

3. In your Methods section, please provide additional details regarding the host plants used in your study and ensure you have described the source. For more information regarding PLOS' policy on materials sharing and reporting, see https://journals.plos.org/plosone/s/materials-and-software-sharing#loc-sharing-materials.

"No"

Reviewers' comments:

Reviewer's Responses to Questions

**Comments to the Author**

1. Is the manuscript technically sound, and do the data support the conclusions?

Reviewer #1: Yes

Reviewer #2: Partly

Reviewer #3: Yes

2. Has the statistical analysis been performed appropriately and rigorously? 

Reviewer #1: Yes

Reviewer #2: I Don't Know

Reviewer #3: Yes

3. Have the authors made all data underlying the findings in their manuscript fully available?

Reviewer #1: Yes

Reviewer #2: Yes

Reviewer #3: Yes

4. Is the manuscript presented in an intelligible fashion and written in standard English?

Reviewer #1: Yes

Reviewer #2: Yes

Reviewer #3: Yes

5. Review Comments to the Author

Reviewer #1: The manuscript number PONE-D-20-28992 entitled « Comparison of life history characteristics of two different genetic clusters of Bemisia tabaci MED (Hemiptera: Aleyrodidae)» has been reviewed.

In this research, authors compared life history parameters of two different genetic clusters of B. tabaci MED (C1 and C2) occurring in Korea on two different host plants (cucumber and tobacco) through single and cross mating. They concluded that the rapid convergence of genetic clusters of B. tabaci MED populations was clearly associated with differences in their life history characteristics. Overall, this is an original, scientifically sound manuscript, adopting correct methodology and adequate writing way with standard English language, with relevant scientific literature used throughout although the DISCUSSION section is clearly missing the insertion of some references and « specific » related data. Some necessary revisions (see below) should be made by the authors to reach a level acceptable for publication in PLOS ONE.

L1 (Title): please replace "life history characteristics" with "life history parameters"

L2: for a sufficiently informative title, please add "through single and cross mating" after "(Hemiptera: Aleyrodidae)"

L16-17: change "is one of serious insect pests with economic importance worldwide" to "is an economically important insect pest worldwide"

L18-19: change "on greenhouse tomatoes" to "occurring in greenhouse tomatoes"

L20: one-year period

L21: replace (here and wherever possible throughout the manuscript) "characteristics" with "parameters"

L22: please delete "of B. tabaci MED" as you already stated this earlier

L41: please specify which kind of "regions" ? (cultivated with which species of host crops); or you simply may change to "agricultural-producing regions"

L41: add a comma before "whereas"

L53: replace "Especially" with "More specifically"

L55: change "to find evidence" with "to provide more evidence"

L79: replace "These two plants belong to the" with "These two species are among the"

L82: Bemisia tabaci colonies were ........ (always write species in full when starting a new sentence)

L92: (< 12-h-old)

L98: before the experiments were initiated.

Page 13 - TABLE 1 (third column): Sing cluster ?? did you mean "Single cluster" ?

L150: A two-way analysis of variance (ANOVA) was

L153: delete "(SAS institute, 2013)" (this is already mentioned in the reference [33])

L153-154: replace the second "(SAS Institute, 2013)" with "[33]"

L176: "Population parameters were as follows:" ; please add the corresponding reference for these formulas

L193: of each treatment group showed

L197: in each treatment group of

L199: the quality of this Figure 1 should be improved

L226: Means followed

L231: Means followed

L239: Means followed

L244: Means followed

L261: Means followed

L267: in the DISCUSSION section, there is a clear lack of info supported with references (references are used only in the first paragraph !!). Please add more relevant references (and related info from other countries / other B. tabaci biotypes - comparisions with your findings) to other parts of this section that needs to be improved to reach an acceptable level for publication. In this context, please note that what you wrote "Life table parameters of B. tabaci have been previously reported regarding different putative species, host plants, or temperatures [12, 13, 34-38]." is a too general statement, so I'd suggest you to give separately (different sentences) more details linked to this general statement and to the supporting references used here (12, 13, 34-38])

L295-296: "Insecticide resistance might also play a role in the prevalence of genetic cluster 2" ; please add references to support this

The Figure page 30 is of bad quality, so it should be improved for more clarity to the reader

Reviewer #2: COMMENTS FOR AUTHORS

I have gone through manuscript “Comparison of life history characteristics of two different genetic clusters of Bemisia tabaci MED (Hemiptera: Aleyrodidae)” and I believe that is not suitable for publication in PLOS ONE Journal.

I have several concerns :

The title doesn’t match with why the study was performed

The provided abstract does not adequately represent the manuscript

The introduction is poorly written for example in lines 48-49 “Potentially different insecticide resistance of these genetic clusters, if any, might be partly involved in genetic cluster change » authors do not report the two important characteristics concern the harbouring of endosymbionts and the transmission of plant viruses that can modify whitefly feeding behaviour. Consider the two references below:

- Kirk, H., Dorn, S. & Mazzi, D. Molecular genetics and genomics generate new insights into invertebrate pest invasions. Evol. Appl. 6, 842–856 (2013).

- Liu, B. M. et al. Multiple forms of vector manipulation by a plant-infecting virus: Bemisia tabaci and tomato yellow leaf curl virus. J. Virol. 87, 4929–4937 (2013).

The Materials and Methods section.

Table 1 need major work :

-Columns 2 and 5 Treatment and crosses, male and female are represented by m and f and also by ♀ and ♂ symbols

- Column 3 culture type, what does it mean sing cluster?

- Column 4 Mating method, the words single and single crossing are somewhat confusing. The single crossing should be replaced by cross mating.

Lastly, the results and discussion sections seem more like a thesis-style

The discussion needs major rewriting to improve clarity indeed authors gave only 4 references (line 271). As mentioned earlier, significant number of questions left unanswered such the role of endosymbiotic bacteria and virus transmission in the whitefly behavior.

Reviewer #3: The manuscript reports an interesting research, which provides light on genetic dynamics in two populations of a species of the Bemisia tabaci group having a relevant applied and economic interest in vast geographical areas. The studies have been adequately structured and planned in the methodology, producing results consistent with the experiment and conclusions adequate to the results obtained. Therefore, the paper is worth of being published on PlosOne, after only very minor changes. In particular, in addition to detecting a few small weakness regarding literature references in some parts of the paper, I am pointing out an aspect that makes me quite dubious and to which I would like to draw the Author's attention (see "main note" in the attached file).

6. PLOS authors have the option to publish the peer review history of their article (what does this mean?). If published, this will include your full peer review and any attached files.

Reviewer #1: No

Reviewer #2: No

Reviewer #3: No

---

## [Author Response · Author response to Decision Letter 0]

14 Jan 2021

We are grateful for the reviewer’s valuable comments to improve our manuscript. We improved our manuscript significantly by incorporating the reviewer’s comments and suggestions. We have colored revised parts in red in the manuscript. We revised some parts of the manuscript more adequately to represent the manuscript than before.

---

## [Decision Letter · Decision Letter 1]

17 Feb 2021

PONE-D-20-28992R1

Comparison of life history parameters of two different genetic clusters of Bemisia tabaci MED (Hemiptera: Aleyrodidae) through single and cross mating

PLOS ONE

Dear Dr. Lee,

Thank you for submitting your manuscript to PLOS ONE. After careful consideration, we feel that it has merit but does not fully meet PLOS ONE’s publication criteria as it currently stands. Therefore, we invite you to submit a revised version of the manuscript that addresses the points raised during the review process.

The scientific quality and general presentation of the manuscript (R1) have been significantly improved following reviewers' comments and suggestions on the original version, nevertheless I've noticed that the authors ignored one of the major comments of the Editor regarding the language (Reminder as stated in the previous decision letter: We suggest you thoroughly copyedit your manuscript for language usage, spelling, and grammar. If you do not know anyone who can help you do this, you may wish to consider employing a professional scientific editing service). Therefore it is mandatory to proofread the manuscript by a native English speaker before to submit the R2 version to consider the manuscript for publication. Also, authors should make some few additional revisions in the MATERIALS AND METHODS and DISCUSSION sections as suggested by the Reviewer #3. All reviewers' comments are stated below.

We look forward to receiving your revised manuscript.

Kind regards,

Ramzi Mansour

Academic Editor

PLOS ONE

Reviewers' comments:

Reviewer's Responses to Questions

**Comments to the Author**

1. If the authors have adequately addressed your comments raised in a previous round of review and you feel that this manuscript is now acceptable for publication, you may indicate that here to bypass the “Comments to the Author” section, enter your conflict of interest statement in the “Confidential to Editor” section, and submit your "Accept" recommendation.

Reviewer #2: All comments have been addressed

Reviewer #3: All comments have been addressed

Reviewer #4: All comments have been addressed

2. Is the manuscript technically sound, and do the data support the conclusions?

Reviewer #2: Yes

Reviewer #3: Yes

Reviewer #4: Yes

3. Has the statistical analysis been performed appropriately and rigorously? 

Reviewer #2: Yes

Reviewer #3: Yes

Reviewer #4: Yes

4. Have the authors made all data underlying the findings in their manuscript fully available?

Reviewer #2: Yes

Reviewer #3: Yes

Reviewer #4: Yes

5. Is the manuscript presented in an intelligible fashion and written in standard English?

Reviewer #2: Yes

Reviewer #3: Yes

Reviewer #4: Yes

6. Review Comments to the Author

Reviewer #2: I think the MS can be accepted for publication in PlosOne journal as authors provided a point-by-point response to my specific comments.

Reviewer #3: VERY MINOR CHANGES STILL REQUIRED:

Line 19: please change “displacement” with “have been displaced”.

Line 46: please change “displacement” with “have been displaced”.

Line 283: please change “displacement of” with “displacement to”.

Reviewer #4: This manuscript compares the life cycle parameters of two different genetic clusters (one dominant the other) of the Bemisia tabaci MED species; it is very clear and well written.

I have only a few comments concerning the genetic analysis:

- In your section of materials and methods for population genetic analysis using STRUCTURE software, I would like to know what model did you use regarding populations mixtures? (ie. “No admixture” or “admixture”). Please indicate this in your M and M section.

- You took only “females” to perform the genetic analysis and, as a result, I expected to see only “hybrid patterns” in your “C1C1 crossings” or at least some, especially with respect to the observed “purity” of your parental lines (at least from C1 on both host plants) in the Bayesian results. I was very surprised to see that each population seems to be so different from the other, looking like different “species” (ie. almost no gene flow could be seen between individuals) despite the fact that in your crossings you obtained hybrids from both populations (biology experiment). You need to comment that in your discussion section and explain why.

- Since you’re using population genetics tools, one would expect that you would also present in the sup data, at least, the basic genetic indeces: allelic richness (na), expected/observed heterozygosity (He/Ho), and the fixation index (Fis) for your tested populations. Indeed, it would be interesting to see the degree of homozygosity obtained in your pure parental populations (even in the small sample you took).

7. PLOS authors have the option to publish the peer review history of their article (what does this mean?). If published, this will include your full peer review and any attached files.

Reviewer #2: No

Reviewer #3: No

Reviewer #4: No

---

## [Author Response · Author response to Decision Letter 1]

4 Mar 2021

Thank you for the comments on our study.

---

## [Editor Report · Decision Letter 2]

8 Mar 2021

Comparison of life history parameters of two different genetic clusters of Bemisia tabaci MED (Hemiptera: Aleyrodidae) through single and cross mating

PONE-D-20-28992R2

Dear Dr. Lee,

We’re pleased to inform you that your manuscript has been judged scientifically suitable for publication and will be formally accepted for publication once it meets all outstanding technical requirements.

Kind regards,

Ramzi Mansour

Academic Editor

PLOS ONE
---

## [Editor Report · Acceptance letter]

12 Mar 2021

PONE-D-20-28992R2 

Comparison of life history parameters of two different genetic clusters of *Bemisia tabaci* MED (Hemiptera: Aleyrodidae) through single and cross mating 

Dear Dr. Lee:

I'm pleased to inform you that your manuscript has been deemed suitable for publication in PLOS ONE. Congratulations! Your manuscript is now with our production department. 

Kind regards, 

on behalf of

Dr. Ramzi Mansour 

Academic Editor

PLOS ONE